# Learning a Neural Semantic Parser from User Feedback

## Abstract

We present an approach to rapidly and easily build natural language interfaces to databases for new domains, whose performance improves over time based on user feedback, and requires minimal intervention. To achieve this, we adapt neural sequence models to map utterances directly to SQL with its full expressivity, bypassing any intermediate meaning representations. These models are immediately deployed online to solicit feedback from real users to flag incorrect queries. Finally, the popularity of SQL facilitates gathering annotations for incorrect predictions using the crowd, which is directly used to improve our models. This complete feedback loop, without intermediate representations or database specific engineering, opens up new ways of building high quality semantic parsers. Experiments suggest that this approach can be deployed quickly for any new target domain, as we show by learning a semantic parser for an online academic database from scratch.

```
Most recent papers of Michael I. Jordan

SELECT paper.paperId, paper.year
FROM paper, writes, author
WHERE paper.paperId = writes.paperId
  AND writes.authorId = author.authorId
  AND author.authorName = "michael i. jordan"
  AND paper.year =
    (SELECT max(paper.year)
     FROM paper, writes, author
     WHERE paper.paperId = writes.paperId
       AND writes.authorId = author.authorId
       AND author.authorName = "michael i. jordan");

I'd like to book a flight from San Diego to Toronto

SELECT DISTINCT f1.flight_id
FROM flight f1, airport_service a1, city c1,
  airport_service a2, city c2
WHERE f1.from_airport = a1.airport_code
  AND a1.city_code = c1.city_code
  AND c1.city_name = 'san diego'
  AND f1.to_airport = a2.airport_code
  AND a2.city_code = c2.city_code
  AND c2.city_name = 'toronto';
```

Figure 1: Utterances with corresponding SQL queries to answer them for two domains, an academic database and a flight reservation database.

## 1 Introduction

Existing semantic parsing approaches for building natural language interfaces to databases (NLIDBs) either use special-purpose intermediate meaning representations that lack the full expressivity of database query languages or require extensive feature engineering, making it difficult to deploy them in new domains. We present a robust approach to quickly and easily learn and deploy semantic parsers from scratch, whose performance improves over time based on user feedback and requires minimal intervention.

To learn these semantic parsers, we (1) adapt neural sequence models to map utterances directly to SQL thereby bypassing intermediate representations and taking full advantage of SQL's querying capabilities, (2) immediately deploy the model online to solicit questions and user feedback on results to reduce SQL annotation efforts, and (3) use crowd workers from skilled markets to provide SQL annotations that can directly be used for model improvement, in addition to being easier and cheaper to obtain than logical meaning representations. We demonstrate the effectiveness of the complete approach by successfully learning a semantic parser for an academic domain by simply deploying it online for three days.

This type of interactive learning is related to a number of recent ideas in semantic parsing, including batch learning of models that directly produce programs (e.g. regular expressions (Locascio et al., 2016)), learning from paraphrases (often gathered through crowdsourcing (Wang et al.,

2015)), data augmentation (e.g. based on manually engineered semantic grammars (Jia and Liang, 2016)) and learning through direct interaction with users (e.g. where a single user teaches the model new concepts (Wang et al., 2016)). However, there are unique advantages to our approach, including showing (1) that non-linguists can write SQL to encode complex, compositional computations (e.g. see Fig 1),[1] (2) that external paraphrase resources and the structure of facts from the target database itself can be used for effective data augmentation, and (3) that actual database users can effectively drive the overall learning by simply providing feedback about what the model is currently getting correct.

Our experiments measure the performance of these learning advances, both in batch on existing datasets and through a simple online experiment for the full interactive setting. For the batch evaluation, we use sentences from the benchmark Geo-Query and ATIS domains, converted to contain SQL meaning representations. Our neural learning with data augmentation achieves near state-of-the-art accuracies, despite the extra complexities of mapping directly to SQL. We also perform simulated interactive learning on this data, showing that with perfect user feedback our full approach could learn high quality parsers with only 55% of the data. Finally, we do a small scale online experiment for a new domain, academic paper metadata search, demonstrating that actual users can provide useful feedback and our full approach is an effective method for learning a high quality parser that continues to improve over time as it is used.

## 2 Related Work

Although diverse meaning representation languages have been used with semantic parsers – such as regular expressions (Kushman and Barzilay, 2013; Locascio et al., 2016) and systems of equations (Kushman et al., 2014; Roy et al., 2016) – parsers for querying databases have typically used either logic programs (Zelle and Mooney, 1996), lambda calculus (Zettlemoyer and Collins,

---

[1] Parsers can also be learned directly from question-answer pairs (Liang et al., 2011). However, recent work has shown that in many domains, especially large databases, it is easier to write a query than to manually sort through the data and list the answers (Yih et al., 2016). Learning parsers directly from SQL queries has the added benefit that we can potentially hire programmers on skilled-labor crowd markets, such as UpWork, to further improve scalability, an exploration we leave to future work.

2005), or $\lambda$-DCS (Liang et al., 2011) as the meaning representation language. All three of these languages are modeled after natural language to simplify parsing. However, none of them is used to query databases outside of the semantic parsing literature; therefore, they are understood by few people and not supported by standard database implementations. In contrast, we parse directly to SQL, which is a popular database query language with wide usage and support.

A few systems have been developed to directly generate SQL queries from natural language (Popescu et al., 2003; Giordani and Moschitti, 2012; Poon, 2013). However, all of these systems make strong assumptions on the structure of queries: they use manually engineered rules that can only generate a subset of SQL, require lexical matches between question tokens and table/column names, or require questions to have a certain syntactic structure. In contrast, our approach can generate arbitrary SQL queries, only uses lexical matching for entity names, and does not depend on syntactic parsing.

We use a neural sequence-to-sequence model to directly generate SQL queries from natural language questions. This approach builds on recent work demonstrating that such models are effective for tasks such as machine translation (Bahdanau et al., 2014) and natural language generation (Kiddon et al., 2016). Recently, neural models have been successfully applied to semantic parsing with simpler meaning representation languages (Dong and Lapata, 2016; Jia and Liang, 2016) and short regular expressions (Locascio et al., 2016). Our work extends these results to the task of SQL generation. Finally, Ling et al. (2016) generate Java/Python code for trading cards given a natural language description; however, this system suffers from low overall accuracy.

A final direction of related work studies methods for reducing the annotation effort required to train a semantic parser. Semantic parsers have been trained from various kinds of annotations, including labeled queries (Zelle and Mooney, 1996; Wong and Mooney, 2007; Zettlemoyer and Collins, 2005), question/answer pairs (Liang et al., 2011; Berant et al., 2013), distant supervision (Krishnamurthy and Mitchell, 2012; Choi et al., 2015), and binary correct/incorrect feedback signals (Clarke et al., 2010). Each of these schemes presents a particular trade-off between annotation

effort and parser accuracy; however, recent work has suggested that labeled queries are the most effective (Yih et al., 2016). Our approach trains on fully labeled SQL queries to maximize accuracy, but uses binary feedback from users to reduce the number of queries that need to be labeled. Annotation effort can also be reduced by using crowd workers to paraphrase automatically generated questions (Wang et al., 2015); however, this approach may not generate the questions that users actually want to ask the database – an experiment in this paper demonstrated that 48% of users' questions in a calendar domain could not be generated.

## 3 Feedback-based Learning

Our feedback-based learning approach can be used to quickly deploy semantic parsers to create NLIDBs for any new domain. Is is a simple interactive learning algorithm that deploys a preliminary semantic parser, then iteratively improves this parser using user feedback and selective query annotation. A key requirement of this algorithm is the ability to cheaply and efficiently annotate queries for chosen user utterances. We address this requirement by developing a model that directly outputs SQL queries (Section 4), which can also be produced by crowd workers.

Our algorithm alternates between stages of training the model and making predictions to gather user feedback, with the goal of improving performance in each successive stage. The procedure is described in Algorithm 1. Our neural model $\mathcal{N}$ is initially trained on synthetic data $T$ generated by domain-independent schema templates (see Section 4), and is then ready to answer new user questions, $n$. The results $\mathcal{R}$ of executing the predicted SQL query $q$ are presented to the user who provides a binary correct/incorrect feedback signal. If the user marks the result correct, the pair $(n, q)$ is added to the training set. If the user marks the result incorrect, the algorithm asks a crowd worker to annotate the utterance with the correct query, $\hat{q}$, and adds $(n, \hat{q})$ to the training set. This procedure can be repeated indefinitely, ideally increasing parser accuracy and requesting fewer annotations in each successive stage.

## 4 Semantic Parsing to SQL

We use a neural sequence-to-sequence model for mapping natural language questions directly

---

**Algorithm 1** Feedback-based learning

1: **procedure** LEARN(schema)
2:     $T \leftarrow$ initial_data(*schema*)
3:     **while** true **do**
4:         $\mathcal{T} \leftarrow T \cup$ paraphrase$(T)$
5:         $\mathcal{N} \leftarrow$ train_model$(\mathcal{T})$
6:         **for each** $n \in$ new utterances **do**
7:             $q \leftarrow$ predict$(\mathcal{N}, n)$
8:             $\mathcal{R} \leftarrow$ execute$(q)$
9:             $f \leftarrow$ feedback$(\mathcal{R})$
10:             **if** $f =$ correct **then**
11:                 $T \leftarrow T \cup (n, q)$
12:             **else if** $f =$ wrong **then**
13:                 $\hat{q} \leftarrow$ annotate$(n)$
14:                 $T \leftarrow T \cup (n, \hat{q})$

---

to SQL queries and this allows us to scale our feedback-based learning approach, by easily crowdsourcing labels when necessary. We further present two data augmentation techniques which use content from the database schema and external paraphrase resources.

### 4.1 Model

We use an encoder-decoder model with global attention, similar to Luong et al. (2015), where the anonymized utterance is encoded using a bidirectional LSTM network, then decoded to directly predict SQL query tokens. Fixed pre-trained word embeddings from word2vec (Mikolov et al., 2013) are concatenated to the embeddings that are learned for source tokens from the training data. The decoder predicts a conditional probability distribution over possible values for the next SQL token given the previous tokens using a combination of the previous SQL token embedding, attention over the hidden states of the encoder network, and an attention signal from the previous time step.

Formally, if $\mathbf{q_i}$ represents an embedding for the $i^{th}$ SQL token $q_i$, the decoder distribution is

$$p(q_i|q_1, \ldots, q_{i-1}) \propto \exp\left(\mathbf{W} \tanh(\hat{\mathbf{W}}[\mathbf{h_i} : \mathbf{c_i}])\right)$$

where $\mathbf{h_i}$ represents the hidden state output of the decoder LSTM at the $i^{th}$ timestep, $\mathbf{c_i}$ represents the context vector generated using an attention weighted sum of encoder hidden states based on $\mathbf{h_i}$, and, $\mathbf{W}$ and $\hat{\mathbf{W}}$ are linear transformations. If $\mathbf{s_j}$ is the hidden representation generated by the encoder for the $j^{th}$ word in the utterance ($k$ words

long), then the context vectors are defined to be

$$\mathbf{c_i} = \sum_{j=1}^{k} \alpha_{i,j} \cdot \mathbf{s_j}$$

The attention weights $\alpha_{i,j}$ are computed using an inner product between the decoder hidden state for the current timestep $\mathbf{h_i}$, and the hidden representation of the $j^{th}$ source token $\mathbf{s_j}$

$$\alpha_{i,j} = \frac{\exp(\mathbf{h_i}^T \mathbf{F} \mathbf{s_j})}{\sum_{j=1}^{k} \exp(\mathbf{h_i}^T \mathbf{F} \mathbf{s_j})}$$

where $\mathbf{F}$ is a linear transformation. The decoder LSTM cell $f$ computes the next hidden state $\mathbf{h_i}$, and cell state, $\mathbf{m_i}$, based on the previous hidden and cell states, $\mathbf{h_{i-1}}, \mathbf{m_{i-1}}$, the embeddings of the previous SQL token $\mathbf{q_{i-1}}$ and the context vector of the previous timestep, $\mathbf{c_{i-1}}$

$$\mathbf{h_i}, \mathbf{m_i} = f(\mathbf{h_{i-1}}, \mathbf{m_{i-1}}, \mathbf{q_{i-1}}, \mathbf{c_{i-1}})$$

We apply dropout on non-recurrent connections for regularization, as suggested by Zaremba et al. (2014). Beam search is used for decoding the SQL queries after learning.

## 4.2 Entity Anonymization

We handle entities in the utterances and SQL by replacing them with their types, using incremental numbering to model multiple entities of the same type (e.g., CITY_NAME_1). During training, when the SQL is available, we infer the type from the associated column name; for example, Boston is a city in city.city_name = 'Boston'. To recognize entities in the utterances at test time, we build a search engine on all entities from the target database. For every span of words (starting with a high span size and progressively reducing it), we query the search engine using a TF-IDF scheme to retrieve the entity that most closely matches the span, then replace the span with the entity's type. We store these mappings and apply them to the generated SQL to fill in the entity names. TF-IDF matching allows some flexibility in matching entity names in utterances, for example, a user could say *Donald Knuth* instead of *Donald E. Knuth*.

## 4.3 Data Augmentation

We present two data augmentation strategies that either (1) provide the initial training data to start the interactive learning, before more labeled examples become available, or (2) use external paraphrase resources to improve generalization.

```
Get all <ENT1>.<NAME> having
  <ENT2>.<COL1>.<NAME> as <ENT2>.<COL1>.<TYPE>

SELECT <ENT1>.<DEF> FROM JOIN_FROM(<ENT1>, <ENT2>)
WHERE JOIN_WHERE(<ENT1>, <ENT2>) AND
  <ENT2>.<COL1> = <ENT2>.<COL1>.<TYPE>
```

(a) Schema template

```
Get all author having dataset  as DATASET_TYPE

SELECT author.authorId
FROM author , writes , paper , paperDataset , dataset
WHERE author.authorId = writes.authorId
  AND writes.paperId = paper.paperId
  AND paper.paperId = paperDataset.paperId
  AND paperDataset.datasetId = dataset.datasetId
  AND dataset.datasetName = DATASET_TYPE
```

(b) Generated utterance-SQL pair

Figure 2: (a) Example schema template consisting of a question and SQL query with slots to be filled with database entities, columns, and values; (b) Entity-anonymized training example generated by applying the template to an academic database.

**Schema Templates**  To bootstrap the model to answer simple questions initially, we defined 22 language/SQL templates that are schema-agnostic, so they can be applied to any database. These templates contain slots whose values are populated given a database schema. An example template is shown in Figure 2a. The <ENT> types represent tables in the database schema, <ENT>.<COL> represents a column in the particular table and <ENT>.<COL>.<TYPE> represents the type associated with the particular column. A template is instantiated by first choosing the entities and attributes. Next, join conditions, i.e., JOIN_FROM and JOIN_WHERE clauses, are generated from the tables on the shortest path between the chosen tables in the database schema graph, which connects tables (graph nodes) using foreign key constraints. Figure 2b shows an instantiation of a template using the path author - writes - paper - paperdataset - dataset. SQL queries generated in this manner are guaranteed to be executable on the target database. On the language side, an English name of each entity is plugged into the template to generate an utterance for the query.

**Paraphrasing**  The second data augmentation strategy uses the Paraphrase Database (PPDB) (Ganitkevitch et al., 2013) to automatically generate paraphrases of training utterances. Such methods have been recently used to improve performance for parsing to logical forms (Chen et al., 2016). PPDB contains over 220 million paraphrase pairs divided into 6 sets (small to XXXL)

based on precision of the paraphrases. We use the one-one and one-many paraphrases from the large version of PPDB. To paraphrase a training utterance, we pick a random word in the utterance that is not a stop word and replace it with a random paraphrase. We perform paraphrase expansion on all examples labeled during learning, as well as the initial seed examples from schema templates.

## 5  Benchmark Experiments

Our first set of experiments demonstrates that our semantic parsing model has comparable accuracy to previous work, despite the increased difficulty of directly producing SQL. We demonstrate this result by running our model on two benchmark datasets for semantic parsing, GEO880 and ATIS.

### 5.1  Data sets

GEO880 is a collection of 880 utterances issued to a database of US geographical facts (Geobase), originally in Prolog format. Popescu et al. (2003) created a relational database schema for Geobase together with SQL queries for a subset of 700 utterances. To compare against prior work on the full corpus, we annotated the remaining utterances and used the standard 600/280 training/test split (Zettlemoyer and Collins, 2005).

ATIS is a collection of 5,418 utterances to a flight booking system, accompanied by a relational database and SQL queries to answer the questions. We use 4,473 utterances for training, 497 for development and 448 for test, following Kwiatkowski et al. (2011). The original SQL queries were very inefficient to execute due to the use of IN clauses, so we converted them to joins (Ramakrishnan and Gehrke, 2003) while verifying that the output of the queries was unchanged.

Table 1 shows characteristics of both data sets. GEO880 has shorter queries but is more compositional: almost 40% of the SQL queries have at least one nested subquery. ATIS has the longest utterances and queries, with an average utterance length of 11 words and an average SQL query length of 67 tokens. They also operate on approximately 6 tables per query on average. We will release our processed versions of both datasets.

### 5.2  Experimental Methodology

We follow a standard train/dev/test methodology for our experiments. The training set is augmented using schema templates and 3 paraphrases per

|  | Geo880 | ATIS | SCHOLAR |
|---|---|---|---|
| Avg. NL length | 7.56 | 10.97 | 6.69 |
| NL vocab size | 151 | 808 | 303 |
| Avg. SQL length | 16.06 | 67.01 | 28.85 |
| SQL vocab size | 89 | 605 | 163 |
| # Subqueries > 1 | 39.8 | 12.42 | 2.58 |
| # Tables | 1.19 | 5.88 | 3.33 |

Table 1: Utterance and SQL query statistics for each dataset. Vocabulary sizes are counted after entity anonymization.

training example, as described in Section 4. Utterances were anonymized by replacing them with their corresponding types and all words that occur only once were replaced by UNK symbols. The development set is used for hyperparameter tuning and early stopping. For GEO880, we use cross validation on the training set to tune hyperparameters. We used a minibatch size of 100 and used Adam (Kingma and Ba, 2014) with a learning rate of 0.001 for 70 epochs for all our experiments. We used a beam size of 5 for decoding. We report test set accuracy of our SQL query predictions by executing them on the target database and comparing the result with the true result.

### 5.3  Results

Tables 2 and 3 show test accuracies based on denotations for our model on GEO880 and ATIS respectively, compared with previous work.[2] To our knowledge, this is the first result on directly parsing to SQL to achieve comparable performance to prior work without using any database-specific feature engineering. Popescu et al. (2003) and Giordani and Moschitti (2012) also directly produce SQL queries but on a subset of 700 examples from GEO880. The former only works on semantically tractable utterances where words can be unambiguously mapped to schema elements, while the latter uses a reranking approach that also limits the complexity of SQL queries that can be handled. GUSP (Poon, 2013) creates an intermediate representation that is then deterministically converted to SQL to obtain an accuracy of $74.8\%$ on ATIS, which is boosted to 83.5% using manually introduced disambiguation rules. However, it requires a lot of SQL specific engineering (for example, special nodes for argmax) and is hard to extend to more complex SQL queries.

On both datasets, our SQL model has slightly

---

[2]Note that 2.8% of GEO880 and 5% ATIS gold test set SQL queries (before any processing) produced empty results.

| System | Acc. |
|---|---|
| Ours (SQL) | 82.5 |
| Popescu et al. (2003) (SQL) | 77.5* |
| Giordani and Moschitti (2012) (SQL) | 87.2* |
| Dong and Lapata (2016) | 84.6◇† |
| Jia and Liang (2016) | 89.3◇ |
| Liang et al. (2011) | 91.1◇ |

Table 2: Accuracy of SQL query results on the Geo880 corpus; * use Geo700; ◇ convert to logical forms instead of SQL; † measure accuracy in terms of obtaining the correct logical form, other systems use denotations

lower accuracy than the best non-SQL results. Most relevant to this work are the neural sequence based approaches of Dong and Lapata (2016) and Jia and Liang (2016). We note that Jia and Liang (2016) use a data recombination technique that boosts accuracy from 85.0 on GEO880 and 76.3 on ATIS; this technique is also compatible with our model. Our results demonstrate that these models are powerful enough to directly produce SQL queries. Thus, our methods enable us to utilize the full expressivity of the SQL language without any extensions that certain logical representations require to answer more complex queries. More importantly, it can be immediately deployed for users in new domains, with a large programming community available for annotation, and thus, fits effectively into a framework for interactive learning.

We perform ablation studies on the development sets (see Table 4) and find that paraphrasing using PPDB consistently helps boost performance. However, unlike in the interactive experiments (Section 6), data augmentation using schema templates does not improve performance in the fully supervised setting.

## 6 Interactive Learning Experiments

In this section, we learn a semantic parser for an academic domain from scratch by deploying an online system using our interactive learning algorithm (Section 3). After three train-deploy cycles, the system correctly answered 63.51% of user's questions. To our knowledge, this is the first effort to learn a semantic parser using a live system, and is enabled by our models that can directly parse language to SQL without manual intervention.

| System | Acc. |
|---|---|
| Ours (SQL) | 79.5 |
| GUSP (Poon, 2013) (SQL) | 74.8 |
| GUSP++ (Poon, 2013) (SQL) | 83.5 |
| Zettlemoyer and Collins (2007) | 84.6◇† |
| Dong and Lapata (2016) | 84.2◇† |
| Jia and Liang (2016) | 83.3◇ |

Table 3: Accuracy of SQL query results on ATIS; ◇ convert to logical forms instead of SQL; † measure accuracy in terms of obtaining the correct logical form, other systems use denotations

| System | GEO880 | ATIS |
|---|---|---|
| Ours | 84.8 | 86.2 |
| - paraphrases | 81.8 | 84.3 |
| - templates | 84.7 | 85.7 |

Table 4: Addition of paraphrases to the training set helps performance, but template based data augmentation does not significantly help in the fully supervised setting. Accuracies reported are cross-validated for Geo880.

### 6.1 User Interface

We developed a web interface for accepting natural language questions to an academic database from users, using our model to generate a SQL query, and displaying the results after execution. Several example utterances are also displayed to help users understand the domain. Together with the results of the generated SQL query, users are prompted to provide feedback which is used for interactive learning. Screenshots of our interface are included in our Supplementary Materials.

Collecting accurate user feedback on predicted queries is a key challenge in the interactive learning setting for two reasons. First, the system's results can be incorrect due to poor entity identification or incompleteness in the database, neither of which are under the semantic parser's control. Second, it can be difficult for users to determine if the presented results are in fact correct. This determination is especially challenging if the system responds with the correct type of result, for example, if the user requests "papers at ACL 2016" and the system responds with all ACL papers.

We address this challenge by providing users with two assists for understanding the system's behavior, and allowing users to provide more granular feedback than simply correct/incorrect.

The first assist is **type highlighting**, which highlights entities identified in the utterance, for example, "paper by *Michael I. Jordan (AUTHOR)* in *ICRA (VENUE)* in *2016 (YEAR).*" This assist is especially helpful because the academic database contains noisy keyword and dataset tables that were automatically extracted from the papers. The second assist is **utterance paraphrasing**, which shows the user another utterance that maps to the same SQL query. For example, for the above query, the system may show "what papers does *Michael I. Jordan (AUTHOR)* have in *ICRA (VENUE)* in *2016 (YEAR).*" This assist only appears if a matching query (after entity anonymization) exists in the model's training set.

Using these assists and the predicted results, users are asked to select from five feedback options: *Correct*, *Wrong Types*, *Incomplete Result*, *Wrong Result* and *Can't Tell*. The *Correct* and *Wrong Result* options represent scenarios when the user is satisfied with the result, or the result is identifiably wrong, respectively. *Wrong Types* indicates incorrect entity identification, which can be determined from type highlighting. *Incomplete Result* indicates that the query is correct but the result is not; this outcome can occur because the database is incomplete. *Can't Tell* indicates that the user is unsure about the feedback to provide.

### 6.2 Three-Stage Online Experiment

In this experiment, using our developed user interface, we use Algorithm 1 to learn a semantic parser from scratch. The experiment had three stages; in each stage, we recruited 10 new users and asked them to issue at least 10 utterances to the system and to provide feedback on the results. We considered results marked as either *Correct* or *Incomplete Result* as correct queries for learning. The remaining incorrect utterances were sent to a crowd worker for annotation and were used to retrain the system for the next stage.

Table 5 shows the percent of utterances judged by users as either *Correct* or *Incomplete Result* in each stage. In the first stage, we do not have any labeled examples, and the model is trained using only synthetically generated data from schema templates and paraphrases (see Section 4.3). Despite the lack of real examples, the system correctly answers 25% of questions. The system's accuracy increases and annotation effort decreases in each successive stage as additional utterances are

| | Stage 1 | Stage 2 | Stage 3 |
|---|---|---|---|
| Accuracy (%) | 25 | 53.7 | 63.5 |

Table 5: Percentage of utterances marked as *Correct* or *Incomplete* by users, in each stage of our online experiment

| Feedback Error Rate (%) | |
|---|---|
| Correct SQL | 6.1 |
| Incorrect SQL | 6.3 |

Table 6: Error rates of user feedback when the SQL is correct and incorrect. The *Correct* and *Incomplete results* options are erroneous if the SQL query is correct, and vice versa for incorrect queries.

contributed and incorrect utterances are labeled. This result demonstrates that we can successfully build semantic parsers for new domains by using neural models to generate SQL with crowd-sourced annotations driven by user feedback.

We analyzed the feedback signals provided by the users in the final stage of the experiment to measure the quality of feedback. We found that 22.3% of the generated queries did not execute (and hence were incorrect). 6.1% of correctly generated queries were marked wrong by users (see Table 6). This erroneous feedback results in redundant annotation of already correct examples. The main cause of this erroneous feedback was incomplete data for aggregation queries, where users chose *Wrong* instead of *Incomplete*. 6.3% of incorrect queries were erroneously deemed correct by users. It is important that this fraction be low, as these queries become incorrectly-labeled examples in the training set that may contribute to the deterioration of model accuracy over time. This quality of feedback is already sufficient for our neural models to improve with usage, and creating better interfaces to make feedback more accurate is an important task for future work.

### 6.3 SCHOLAR dataset

We release a new semantic parsing dataset for academic database search using the utterances gathered in the user study. We augment these labeled utterances with additional utterances labeled by crowd workers. (Note that these additional utterances were not used in the online experiment). The final dataset comprises 816 natural language utterances labeled with SQL, divided

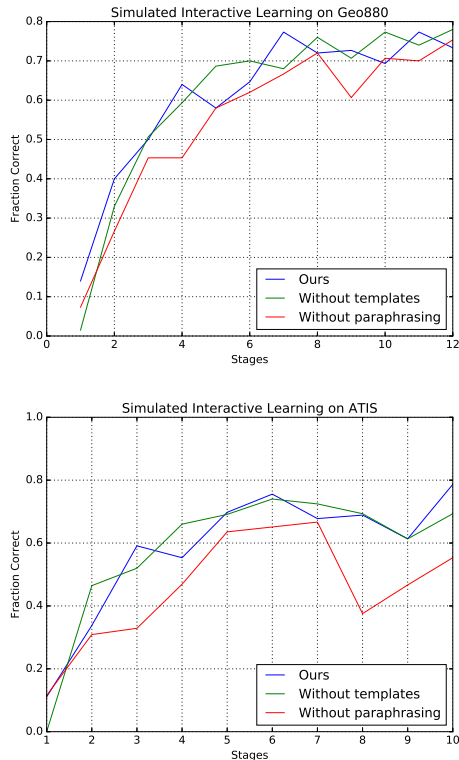

Figure 3: Accuracy as a function of batch number in simulated interactive learning experiments on Geo880 (top) and ATIS (bottom).

into a 600/216 train/test split. We also provide a database on which to execute these queries containing academic papers with their authors, citations, journals, keywords and datasets used. Table 1 shows statistics of this data set. Our parser achieves an accuracy of 67% on this train/test split in the fully supervised setting.

### 6.4   Simulated Interactive Experiments

We conducted additional simulated interactive learning experiments using GEO880 and ATIS to better understand the behavior of our train-deploy feedback loop, the effects of our data augmentation approaches, and the annotation effort required. We randomly divide each training set into $K$ batches and present these batches sequentially to our interactive learning algorithm. Correctness feedback is provided by comparing the result of the predicted query to the gold query, i.e., we assume that users are able to perfectly distinguish correct results from incorrect ones.

Figure 3 shows accuracies on GEO880 and ATIS respectively of each batch when the model is trained on all previous batches. As in the live experiment, accuracy improves with successive batches. Data augmentation using templates helps

| Batch Size | 150 | 100 | 50 |
|---|---|---|---|
| % Wrong | 70.2 | 60.4 | 54.3 |

Table 7: Percentage of examples that required annotation (i.e., where the model initially made an incorrect prediction) on GEO880 vs. batch size.

in the initial stages of GEO880, but its advantage is reduced as more labeled data is obtained. Templates did not improve accuracy on ATIS, possibly because most ATIS queries involve two entities, i.e., a source city and a destination city, whereas our templates only generate questions with a single entity type. Nevertheless, templates are important in a live system to motivate users to interact with it in early stages. As observed before, paraphrasing improves performance at all stages.

Table 7 shows the percent of examples that require annotation using various batch sizes for GEO880. Smaller batch sizes reduce annotation effort, with a batch size of 50 requiring only 54.3% of the examples to be annotated. This result demonstrates that more frequent deployments of improved models leads to fewer mistakes.

## 7   Conclusion

We describe an approach to rapidly train a semantic parser as a NLIDB that iteratively improves parser accuracy over time while requiring minimal intervention. Our approach uses an attention-based neural sequence-to-sequence model, with data augmentation from the target database and paraphrasing, to parse utterances to SQL. This model is deployed in an online system, where user feedback on its predictions is used to select utterances to send for crowd worker annotation.

We find that the semantic parsing model is comparable in performance to previous systems that either map from utterances to logical forms, or generate SQL, on two benchmark datasets, GEO880 and ATIS. We further demonstrate the effectiveness of our online system by learning a semantic parser from scratch for an academic domain. A key advantage of our approach is that it is not language-specific, and can easily be ported to other commonly- used query languages, such as SPARQL or ElasticSearch. Finally, we also release a new dataset of utterances and SQL queries for an academic domain.

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
