# Peer review of "Learning a Neural Semantic Parser from User Feedback"

_ACL 2017 — decision unknown_

[Official Review · Reviewer 1 · rating 4 · confidence 4]
soundness 3 · originality 4 · clarity 5 · impact 4 · substance 4 · appropriateness 5 · meaningful comparison 4 · presentation format Oral Presentation

The paper presents a neural model for predicting SQL queries directly from
natural language utterances, without going through an intermediate formalism.
In addition, an interactive online feedback loop is proposed and tested on a
small scale.

- Strengths:

1\ The paper is very clearly written, properly positioned, and I enjoyed
reading it.

2\ The proposed model is tested and shown to perform well on 3 different
domains (academic, geographic queries, and flight booking)

3\ The online feedback loop is interesting and seems promising, despite of the
small scale of the experiment.

4\ A new semantic corpus is published as part of this work, and additionally
two
existing corpora are converted to SQL format, which I believe would be
beneficial for future work in this area.

- Weaknesses / clarifications:

1\ Section 4.2 (Entity anonymization) - I am not sure I understand the choice
of the length of span for querying the search engine. Why and how is it
progressively reduced? (line 333).

2\ Section 5 (Benchmark experiments) - If I understand correctly, the feedback
loop (algorithm 1) is *not* used for these experiments. If this is indeed the
case, I'm not sure when does data augmentation occur. Is all the annotated
training data augmented with paraphrases? When is the "initial data" from
templates added? Is it also added to the gold training set? If so, I think it's
not surprising that it doesn't help much, as the gold queries may be more
diverse.  In any case, I think this should be stated more clearly. In addition,
I think it's interesting to see what's the performance of the "vanilla" model,
without any augmentation, I think that this is not reported in the paper.

3\ Tables 2 and 3: I find the evaluation metric used here somewhat unclear. 
Does the accuracy measure the correctness of the execution of the query (i.e.,
the retrieved answer) as the text seem to indicate? (Line 471 mentions
*executing* the query). Alternatively, are the queries themselves compared? (as
seems to be the case for Dong and Lapata in Table 2). If this is done
differently for different systems (I.e., Dong and Lapata), how are these
numbers comparable? In addition, the text mentions the SQL model has "slightly
lower accuracy than the best non-SQL results" (Line 515), yet in table 2 the
difference is almost 9 points in accuracy.  What is the observation based upon?
Was some significance test performed? If not, I think the results are still
impressive for direct to SQL parsing, but that the wording should be changed,
as the difference in performance does seem significant.

4\ Line 519 - Regarding the data recombination technique used in Jia and Liang
(2016): Since this technique is applicable in this scenario, why not try it as
well?  Currently it's an open question whether this will actually improve
performance. Is this left as future work, or is there something prohibiting the
use of this technique?

5\ Section 6.2 (Three-stage online experiment) - several details are missing /
unclear:

* What was the technical background of the recruited users?

* Who were the crowd workers, how were they recruited and trained?

* The text says "we recruited 10 new users and asked them to issue at least 10
utterances". Does this mean 10 queries *each* (e.g., 100 overall), or 10 in
total (1 for each).

* What was the size of the initial (synthesized) training  set? 

* Report statistics of the queries - some measure of their lexical variability
/ length / complexity of the generated SQL? This seems especially important for
the first phase, which is doing surprisingly well. Furthermore, since SCHOLAR
uses SQL and NL, it would have been nice if it were attached to this
submission, to allow its review during this period.

6\ Section 6.3 (SCHOLAR dataset)

* The dataset seems pretty small in modern standards (816 utterances in total),
while one of the main advantages of this process is its scalability. What
hindered the creation of a much larger dataset?

* Comparing performance - is it possible to run another baseline on this newly
created dataset to compare against the reported 67% accuracy obtained in this
paper (line 730).

7\ Evaluation of interactive learning experiments (Section 6): I find the
experiments to be somewhat hard to replicate as they involve manual queries of
specific annotators. For example, who's to say if the annotators in the last
phase just asked simpler questions? I realise that this is always problematic
for online learning scenarios, but I think that an effort should be made
towards an objective comparison. For starters, the statistics of the queries
(as I mentioned earlier) is a readily available means to assess whether this
happens. Second, maybe there can be some objective held out test set? This is
problematic as the model relies on the seen queries, but scaling up the
experiment (as I suggested above) might mitigate this risk. Third, is it
possible to assess a different baseline using this online technique? I'm not
sure whether this is applicable given that previous methods were not devised as
online learning methods.

- Minor comments:

1\ Line 48 - "requires" -> "require"

2\ Footnote 1 seems too long to me. Consider moving some of its content to the
body of the text.

3\ Algorithm 1: I'm not sure what "new utterances" refers to (I guess it's new
queries from users?). I think that an accompanying caption to the algorithm
would make the reading easier.

4\ Line 218 - "Is is" -> "It is"

5\ Line 278 mentions an "anonymized" utterance. This confused me at the first
reading, and if I understand correctly it refers to the anonymization described
later in 4.2. I think it would be better to forward reference this. 

- General Discussion:

Overall, I like the paper, and given answers to the questions I raised above,
would like to see it appear in the conference.

- Author Response:

I appreciate the detailed response made by the authors, please include these
details in a final version of the paper.

[Official Review · Reviewer 2 · rating 4 · confidence 4]
soundness 3 · originality 4 · clarity 5 · impact 4 · substance 4 · appropriateness 5 · meaningful comparison 4 · presentation format Oral Presentation

This paper proposes a simple attention-based RNN model for generating SQL
queries from natural language without any intermediate representation. Towards
this end they employ a data augmentation approach where more data is
iteratively collected from crowd annotation, based on user feedback on how well
the SQL queries produced by the model do. Results on both the benchmark and
interactive datasets show that data augmentation is a promising approach.

Strengths:

- No intermediate representations were used. 

- Release of a potentially valuable dataset on Google SCHOLAR.

Weaknesses:

- Claims of being comparable to state of the art when the results on GeoQuery
and
ATIS do not support it. 

General Discussion:

This is a sound work of research and could have future potential in the way
semantic parsing for downstream applications is done. I was a little
disappointed with the claims of “near-state-of-the-art accuracies” on ATIS
and GeoQuery, which doesn’t seem to be the case (8 points difference from
Liang et. al., 2011)). And I do not necessarily think that getting SOTA numbers
should be the focus of the paper, it has its own significant contribution. I
would like to see this paper at ACL provided the authors tone down their
claims, in addition I have some questions for the authors.

- What do the authors mean by minimal intervention? Does it mean minimal human
intervention, because that does not seem to be the case. Does it mean no
intermediate representation? If so, the latter term should be used, being less
ambiguous.

- Table 6: what is the breakdown of the score by correctness and
incompleteness?
What % of incompleteness do these queries exhibit?

- What is expertise required from crowd-workers who produce the correct SQL
queries? 

- It would be helpful to see some analysis of the 48% of user questions which
could not be generated.

- Figure 3 is a little confusing, I could not follow the sharp dips in
performance without paraphrasing around the 8th/9th stages. 

- Table 4 needs a little more clarification, what splits are used for obtaining
the ATIS numbers?

I thank the authors for their response.

[Official Review · Reviewer 3 · rating 4 · confidence 3]
soundness 3 · originality 4 · clarity 4 · impact 4 · substance 3 · appropriateness 5 · meaningful comparison 4 · presentation format Oral Presentation

This paper proposes an approach to learning a semantic parser using an
encoder-decoder neural architecture, with the distinguishing feature that the
semantic output is full SQL queries. The method is evaluated over two standard
datasets (Geo880 and ATIS), as well as a novel dataset relating to document
search.

This is a solid, well executed paper, which takes a relatively well
established technique in the form of an encoder-decoder with some trimmings
(e.g. data augmentation through paraphrasing), and uses it to generate SQL
queries, with the purported advantage that SQL queries are more expressive
than other semantic formalisms commonly used in the literature, and can be
edited by untrained crowd workers (familiar with SQL but not semantic
parsing). I buy that SQL is more expressive than the standard semantic
formalisms, but then again, were there really any queries in any of your three
datasets where the standard formalisms are unable to capture the full
semantics of the query? I.e. are they really the best datasets to showcase the
expressivity of SQL? Also, in terms of what your model learns, what fraction
of SQL does it actually use? I.e. how much of the extra expressivity in SQL is
your model able to capture? Also, does it have biases in terms of the style of
queries that it tends to generate? That is, I wanted to get a better sense of
not just the *potential* of SQL, but the actuality of what your model is able
to capture, and the need for extra expressivity relative to the datasets you
experiment over. Somewhat related to this, at the start of Section 5, you
assert that it's harder to directly produce SQL. You never actually show this,
and this seems to be more a statement of the expressivity of SQL than anything
else (which returns me to the question of how much of SQL is the model
actually generating).

Next, I would really have liked to have seen more discussion of the types of
SQL queries your model generates, esp. for the second part of the evaluation,
over the SCHOLAR dataset. Specifically, when the query is ill-formed, in what
ways is it ill-formed? When a crowd worker is required to post-edit the query,
how much effort does that take them? Equally, how correct are the crowd
workers at constructing SQL queries? Are they always able to construct perfect
queries (experience would suggest that this is a big ask)? In a similar vein
to having more error analysis in the paper, I would have liked to have seen
agreement numbers between annotators, esp. for Incomplete Result queries,
which seems to rely heavily on pre-existing knowledge on the part of the
annotator and therefore be highly subjective.

Overall, what the paper achieves is impressive, and the paper is well
executed; I just wanted to get more insights into the true ability of the
model to generate SQL, and a better sense of what subset of the language it
generates.

A couple of other minor things:

l107: "non-linguists can write SQL" -- why refer to "non-linguists" here? Most
linguists wouldn't be able to write SQL queries either way; I think the point
you are trying to make is simply that "annotators without specific training in
the semantic translation of queries" are able to perform the task

l218: "Is is" -> "It is"

l278: it's not clear what an "anonymized utterance" is at this point of the
paper

l403: am I right in saying that you paraphrase only single words at a time?
Presumably you exclude "entities" from paraphrasing?

l700: introduce a visual variable in terms of line type to differentiate the
three lines, for those viewing in grayscale

There are various inconsistencies in the references, casing issues
(e.g. "freebase", "ccg"), Wang et al. (2016) is missing critical publication
details, and there is an "In In" for Wong and Mooney (2007)